



# Seasonal to interannual variabilities of sea–air CO₂ exchange across Tropical Maritime Continent indicated by eddy–permitting coupled OGCM experiment

Faisal Amri[1], Takashi Nakamura[1], Atsushi Watanabe[2], Aditya. R. Kartadikaria[3], and Kazuo Nadaoka[1]

[1] Transdisciplinary Science and Engineering Department, Tokyo Institute of Technology, Tokyo, Japan
[2] Ocean Policy Research Institute, The Sasakawa Peace Foundation, Japan
[3] Faculty of Earth Sciences and Technology, Oceanography Research Group, Institut Teknologi Bandung, Bandung, Indonesia

*Correspondence to*: Faisal Amri (amri.f.aa@m.titech.ac.jp/faisal.amri.os12@gmail.com)

**Abstract.** The lack of long–term observational data has limited research on sea–air $CO_2$ exchange variabilities in the Tropical Maritime Continent (TMC). To address the issue, we utilized a three–dimensional high–resolution physical–biogeochemical ocean numerical model and applied it to simulate sea–air $CO_2$ exchange in the region over the last decade (2010–2019). Some key features like atmospheric $CO_2$ source signature and high sea surface $pCO_2$ environment inside the TMC were captured by the model. Within the TMC, model results indicated strong $CO_2$ degassing along the south of Java associated with the seasonal cycle of the upwelling system. Abundant supply of inorganic carbon during upwelling season and strong wind speed results in $CO_2$ degassing that could reach as high as 30 gC $m^{-2}$ $year^{-1}$ around the area. In addition to the region acting as a full–year atmospheric $CO_2$ source, the TMC also exhibited interannual modulation in both sea–air $CO_2$ flux and sea surface $pCO_2$ which can be related to the El Niño–Southern Oscillation (ENSO) and Indian Ocean Dipole (IOD). Large–scale anomalous strong $CO_2$ degassing and high sea surface $pCO_2$ from 2015 to 2016 in response to the 2015/2016 El Niño evolution was observed and dominated by modulation within the TMC. It is further found that modulation of $CO_2$ degassing related to IOD were confined along the west of south of Java with a higher magnitude compared with anomalies related to ENSO which shows larger spatial scale but lower in the magnitude. Study conducted here may provide insight about possible variabilities of sea–air $CO_2$ exchange in the area that still poorly represented in many global–scale modelling and reconstruction efforts.



## 1 Introduction

The Tropical Maritime Continent (TMC) region acts as a water passage that allows Pacific Ocean water to be transported to the Indian Ocean as part of thermohaline circulation, which modulates the global climate system (Gordon, 1986; Wyrtki, 1961). Located between the Indian and Pacific Ocean, the TMC area is subject to modulation caused by variabilities occurring in these two ocean basins, including the El Niño–Southern Oscillation (ENSO) in the tropical Pacific Ocean, and Indian Ocean Dipole (IOD) in the Indian Ocean (Ashok et al., 2003; Saji and Yamagata, 2003; Sprintall et al., 2014). Recent

studies have confirmed that these climate modes influence the TMC area through sea–air interaction perturbation, which affects the rainfall rate and oceanic properties such as sea surface temperature, sea surface height, and circulation pattern (Delman et al., 2016; Pujiana et al., 2019, 2020; Saji and Yamagata, 2003; Siswanto et al., 2020; Sprintall et al., 2014; Susanto et al., 2001; Syamsudin et al., 2004).

      Despite the progress, studies on oceanic carbon cycle dynamics in the area remain very limited compared to the number of

oceanic carbon–related studies that are growing globally (Bakker et al., 2016; Key et al., 2004; Takahashi et al., 2002, 2009). Although one of the latest observations–based studies by Kartadikaria et al. (2015) on the compilation of sea surface $CO_2$ partial pressure (sea surface $pCO_2$) across the Indonesian seas could provide a general view of the atmospheric $CO_2$ sink/source characteristics, it still could not represent the actual seasonal cycle and response of the sea–air $CO_2$ exchange to large–scale climate variabilities. Typical sea surface $pCO_2$ underway measurements conducted in a short period are not

reliable in capturing the low–frequency variabilities that usually develop within an interannual time scale or longer (Sutton et al., 2017b). A study by Hamzah et al. (2020) in western Indonesian seas later confirmed this issue by highlighting the possible variation in the carbonate system over seasonal and interannual timescales in the undersampled area. The recent development of empirical model (Iida et al., 2015) or machine learning (Landschützer et al., 2016) in estimating sea surface $pCO_2$ and sea–air $CO_2$ exchange is unfortunately producing a relatively coarse resolution for resolving the complex island

configuration within the TMC and showed inconsistent atmospheric $CO_2$ sink/source signature with observations–based study. These constraints make it challenging to apprehend TMC sea–air $CO_2$ exchange variabilities at various time scales.

      Several modelling studies have indicated that the sea surface $pCO_2$ and sea–air $CO_2$ flux exhibit apparent modulation related to climate variability. A modelling study by Chai et al. (2009) in the South China Sea showed that sea surface $pCO_2$ followed the seasonal variations of net primary productivity, which was inversely correlated with the sea surface temperature

(SST) anomaly in the Eastern Tropical Pacific region (NINO3). Global–scale modelling by Obata and Kitamura (2003) emphasized the Tropical Pacific Ocean sea–air $CO_2$ flux, where the variability in the region related to ENSO contributed approximately 70% to the global variability. Similar global–scale modelling was conducted by Valsala et al. (2014); despite indicating differences in the contribution of Tropical Pacific sea–air $CO_2$ flux variabilities to global variabilities, the study still agrees to the extent that Tropical Pacific Ocean variability has a significant influence on global carbon cycle

modulation. They further suggested a stronger influence of El Niño–Modoki (Ashok et al., 2007) on upper–carbon cycle variability in the western part of the Tropical Pacific, adjacent to the TMC. In their modelling study, Xiu and Chai (2014)





also addressed the significance of the Pacific Decadal Oscillation and North Pacific Gyre Oscillation in modulating the sea–air $CO_2$ flux across the North Pacific region, highlighting the variabilities in much lower frequency domains. These studies partly confirm the hypothesis about the possible low–frequency modulation of the sea–air $CO_2$ exchange, considering the proximity of the TMC to the area studied previously. One of the remaining questions concerns the modulation pattern related to the Indo–Pacific climate variability in the area, which this study attempts to address. Information regarding sea–air $CO_2$ exchange variabilities in the TMC can enhance our understanding about overall atmospheric $CO_2$ sink/source variations across the tropics and its contribution to the global carbon cycle.

A high–resolution coupled Ocean General Circulation Model (OGCM) with low–trophic ecosystem module was employed to further resolve the issue of elucidating the sea–air $CO_2$ exchange variability across the TMC. The model was forced by realistic high–temporal resolution atmospheric forcings to approach the actual ocean–atmosphere dynamics that occurred during the simulation period. We further focused the analysis between 2010 and 2019 to examine the interannual changes in sea surface $pCO_2$ and sea–air $CO_2$ flux in the region. The analysis period included extreme events, such as the 2015/2016 El Niño and the 2019 positive IOD (pIOD). Previous studies have indicated that unprecedented anomalies occur around the TMC associated with these extreme climate events (e.g., Lu and Ren, 2020; Pujiana et al., 2019) and thus, have become an interesting period to examine the sensitivity of the sea–air $CO_2$ exchange variabilities in the area to such anomalous climate events.

## 2 Model and datasets

### 2.1 Model description

The low–trophic ecosystem model employed here was based on carbon (C) and nutrient (phosphate, nitrate, ammonium) tracing, low–trophic ecosystem model developed by Nakamura et al. (2018). The model was embedded in the Coupled Ocean–Atmosphere–Wave and Sediment Transport (COAWST) modelling environment (Warner et al., 2010) with the Regional Ocean Modelling System (ROMS; Shchepetkin and McWilliams, 2005) as the OGCM. Note that although model in Nakamura et al. (2018) mainly focuses on coral reef ecosystem, application on regional scale modelling was possible by deactivating the coral reef model and mainly relies on the low–trophic ecosystem compartment which further modified in this study. The low–trophic ecosystem model includes three phytoplankton functional types (PFT) in terms of carbon biomass, comprising diatoms, dinoflagellates, and coccolithophores. These PFTs utilize nutrients and total dissolved inorganic carbon (DIC) for photosynthesis and assimilation with dissolved oxygen as a by–product. The PFTs were distinguished by different assimilation efficiency, mortality rate, sinking velocity, and survivability under zooplankton grazing. Additionally, coccolithophores PFT use total alkalinity in addition to the DIC for the calcification process to produce $CaCO_3$ shells which explicitly calculated in this model. Details of relevant parametrization used for each PFT in this study can be seen in Table 1.



The material excreted by PFTs following the assimilation process immediately enters the labile dissolved organic matter (labile–DOM) pool. All dead phytoplankton biomass enters the particulate organic matter pool as detritus (detritus–POM)

and immediately sinks into a deeper layer in the water column. As for the dead coccolithophore biomass, the previously produced $CaCO_3$ from the calcification process enters the particulate inorganic matter ($CaCO_3$–PIM) pool and sinks into a deeper layer, like the detritus–POM. Estimated produced $CaCO_3$–PIM from coccolithophore dead biomass was adapted from Krumhardt et al. (2017, 2019) which incorporates the effect of nutrient limitation, water temperature, and dissolved $CO_2$ gas on calcification efficiency.

One type of zooplankton in terms of carbon biomass was assigned in this model which grazed on phytoplankton, labile–DOM, and detritus–POM. As in the phytoplankton, the dead bodies of zooplankton also entered the detritus–POM pool, with a small part entering the $CaCO_3$–PIM pool. The $CaCO_3$–PIM fraction from zooplankton dead biomass was based on Ishizu et al. (2019, 2020).

The decomposition process takes place in the labile–DOM pool to resupply the dissolved inorganic carbon, nitrogen (as

ammonium), and phosphorus (as phosphate) needed by phytoplankton. The decomposition of detritus–POM transforms detritus–POM into labile–DOM and dissolved inorganic material compounds (i.e., DIC, phosphate, and ammonium) simultaneously. Nitrate in this model was recovered through the nitrification of ammonium. We applied the first order dissolution reaction equation for the dissolution process of $CaCO_3$–PIM with a seawater $CaCO_3$ saturation state that varied within the water column (Jansen et al., 2002; Sarmiento and Gruber, 2006). Here, the $CaCO_3$–PIM saturation state was

approximated as the calcite saturation state, given that the main $CaCO_3$ produced in this model came from coccolithophores. Dissolution of $CaCO_3$–PIM resupply the DIC and alkalinity as well. Schematic figure of material flow in our low–trophic ocean ecosystem model can be seen in Figure 1. Details on model formulation and additional parameters used in this model can be seen in the supplementary section 1 (S1).



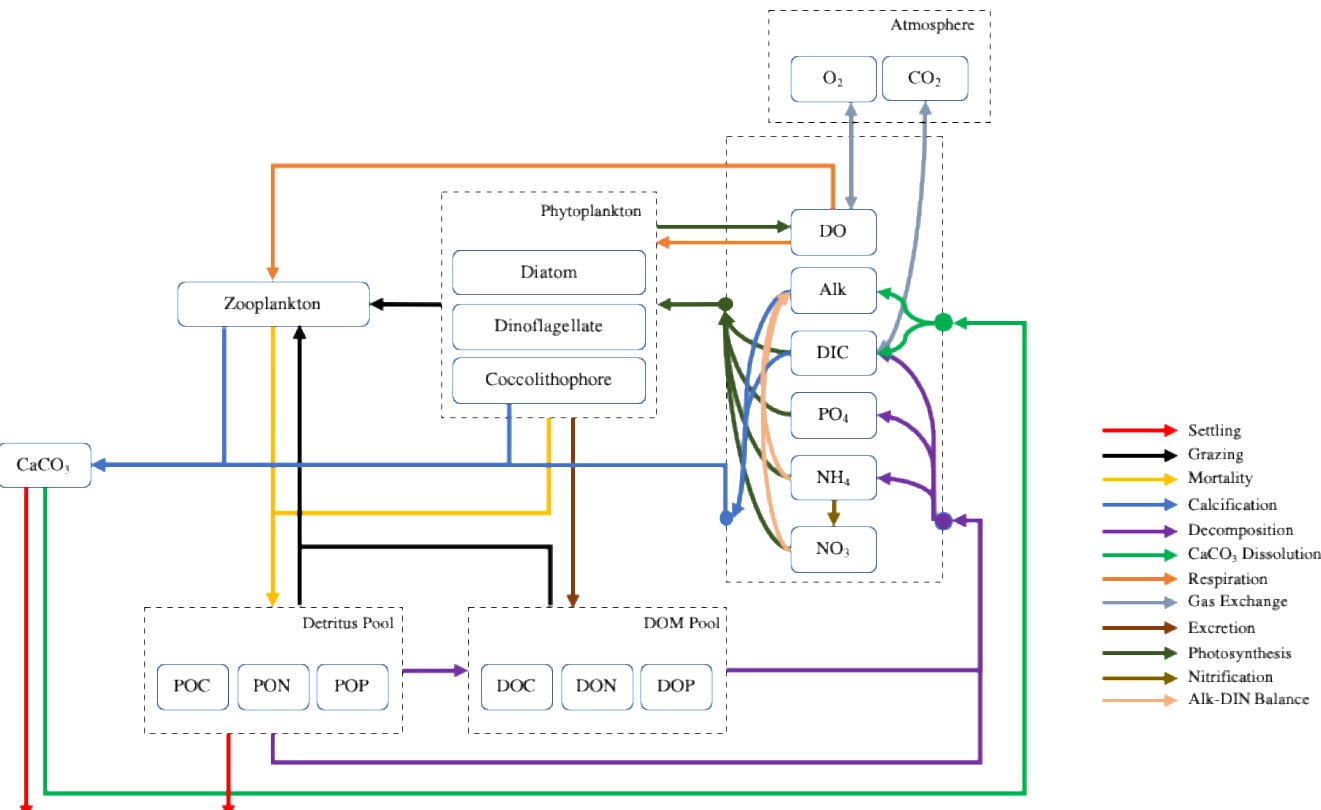

**Figure 1. Schematic figure of low–trophic ecosystem model used in this study.**

Sea surface $pCO_2$ was calculated in each time–step of simulation, set as 120 seconds, using modelled surface layer DIC, total alkalinity, water temperature, and salinity value following OCMIP protocol (Najjar and Orr, 1999). Exchange of $CO_2$ gas between sea surface and atmosphere further calculated by incorporating wind speed and difference between sea surface 120 $pCO_2$ and atmosphere $pCO_2$ as in Wanninkhof (1992) with $CO_2$ solubility parameterization following Weiss (1974). By allowing the gas exchange, the DIC within the sea surface layer is subject to dynamics from sea–air interaction in addition to the biogeochemical processes within the seawater. The sea surface $pCO_2$ can be decomposed into four driving components of sea surface temperature (SST), sea surface salinity (SSS), sea surface total DIC (SSDIC), and sea surface total alkalinity (SSAlk) following Takahashi et al. (1993)


$$dpCO_2 = \left(\frac{\partial pCO_2}{\partial SST}\right)dSST + \left(\frac{\partial pCO_2}{\partial SSS}\right)dSSS + \left(\frac{\partial pCO_2}{\partial SSDIC}\right)dSSDIC + \left(\frac{\partial pCO_2}{\partial SSAlk}\right)dSSAlk$$

We set the model domain to span from the Southeast Tropical Indian Ocean (SETIO) to the Northwest Pacific Ocean (90ºE– 164ºE; 18ºS–29ºN). The domain was gridded uniformly with a horizontal resolution of 1/6° × 1/6° while the water column 130 was transformed into 30–layers of non–uniform, terrain–following s–coordinates. Simulation experiment was started from



December 2007 to January 2020. Results analysed in this study covers the 2010–2019 period considering the first two years simulation as spin–up period. Generic length scale (GLS) mixing parameterizations of the k–ε configuration were utilized in this model for vertical mixing combined with the Kantha–Clayson stability function (CPP options KANTHA_CLAYSON) and horizontal smoothing of buoyancy/shear (CPP options N2S2_HORAVG). Smagorinsky–like diffusion (CPP option
UV_SMAGORINSKY and TS_SMAGORINSKY) was activated in this simulation for the horizontal diffusion and viscosity for both momentum and tracer variables.

**Table 1. Relevant parameterizations for each phytoplankton functional types (PFT) used in simulation experiment. All values shown here were coarsely calibrated from Gregg et al. (2007) and Krumhardt et al. (2019).**

|  | Dinoflagellate | Diatom | Coccolithophore |
|---|---|---|---|
| Maximum photosynthetic rate at 0ºC (day$^{-1}$) | 0.44 | 0.50 | 0.47 |
| Optimum light intensity (J m$^2$ s$^{-1}$) | 87 | 93 | 71 |
| Maximum grazing rate by zooplankton at 0ºC (day$^{-1}$) | 0.36 | 0.34 | 0.29 |
| Threshold value for grazing by zooplankton (µmolC L$^{-1}$) | 0.054 | 0.072 | 0.082 |
| PO$_4$ half saturation constant (µmol L$^{-1}$) | 0.005 | 0.050 | 0.006 |
| NO$_3$ half saturation constant (µmol L$^{-1}$) | 0.20 | 0.50 | 0.20 |
| NH$_4$ half saturation constant (µmol L$^{-1}$) | 0.01 | 0.05 | 0.01 |
| Sinking Velocity (m day$^{-1}$) | 0.25 | 0.75 | 1.00 |

**2.2 Model forcing**

To generate the physical–biogeochemical ocean dynamics within the model domain, the model was by forced three main components consisted of tidal forcing, atmospheric forcing, and atmospheric $CO_2$ concentration. The Oregon State University TPXO tides model output (Egbert and Erofeeva, 2002) and 55–years Japan Reanalysis product (JRA–55; Kobayashi et al., 2015) as tidal forcing and atmospheric forcing, respectively, were used to generate the circulation dynamics
(Physical aspect) in the model domain. Utilized atmospheric forcing here include the three–hourly surface air pressure, air temperature, humidity, wind speed, precipitation, and cloud fraction. Bulk fluxes (i.e., Shortwave radiation, longwave radiation, latent heat, and sensible heat) were computed internally in the model. Annual global–averaged atmospheric $CO_2$ concentration recorded in the NOAA Earth System Research Laboratory (NOAA ESRL) was used and applied uniformly across the entire model domain to generate the $CO_2$ exchange between sea surface and atmosphere (Biogeochemical aspect).




River discharge across the model domain was not implemented in this study; thus, the indicated results of $pCO_2$ and sea–air $CO_2$ flux were caused solely by the ocean–atmosphere interaction dynamics.

## 2.3 Model initialization and boundary condition

The circulation model was initialized using Global Ocean Forecasting System (GOFS) analysis/reanalysis product (Chassignet et al., 2006) which provides horizontal momentum ($u, v, \bar{u}, \bar{v}$), water temperature (T), water salinity (S), and sea
surface height ($\eta$) information. Information in domain's boundary also provided by the same GOFS dataset. Initial and boundary condition for biogeochemical tracers were provided by analytically estimating the vertical profile of phytoplankton, zooplankton, DIC, Alkalinity, nutrients (Nitrate, Ammonium, and Phosphate), and dissolved oxygen. Analytical expression for necessary parameters in the low–trophic ecosystem model was established using Global Data Analysis Project 2nd version (GLODAPv2; Key et al., 2004) which stores scientific cruise data across the globe including
some areas within modelling domain. Each of observed total DIC (Total $CO_2$ in GLODAPv2), total alkalinity, and dissolved oxygen were paired with observed water temperature to create a fourth–order polynomial equation using least–square method. The polynomial equation was later applied to the model's initial and boundary condition by using GOFS water temperature data.

The nutrients in the model's initial and boundary condition further calculated using linear relationship between salinity–
normalized DIC and nutrient concentration indicated by the GLODAPv2 data adapting Sarmiento and Gruber (2006). The linear equation then utilized estimated DIC value and Salinity from the GOFS. We found that the C:P and C:N ratio of 131.9 and 9.2 which higher than Redfield ratio (Redfield, 1934) but still lower than contemporary estimation in Martiny et al. (2014). Observed C:P and C:N values from the GLODAPv2 were later used in the model as well. Through this approach, we could create initial condition in the area with sparse observation record such as the TMC. The initial and boundary condition
for phytoplankton and zooplankton on the other hand, used simpler analytical approach where biomass was calculated as function of vertical position. More details on the analytical equations used to create the initial and boundary conditions of the low–trophic ecosystem model is provided in Table 2.

$$Phy(z) = \frac{10.5 - 0.00095 \times (z + 50)^2}{24} \quad \text{For } z > -155 \text{ m}$$

$$Zoo(z) = 0.1 \times Phy(z)$$

Lateral boundary condition was set to be mixed radiation–nudging for the 3D momentum and tracer variables. The inflow nudging timescale for the temperature/salinity and biogeochemical tracers were set to 100 days and 180 days, respectively.

**Table 2. Analytical equations used to estimate necessary biogeochemical parameters in initial and boundary conditions of the**
**ecosystem model. For application in the simulation experiment, water temperature (T) and salinity (S) in the equations were obtained from the Global Ocean Forecasting System (GOFS) product.**





| Parameter (unit) | Equation |
|---|---|
| Dissolved inorganic carbon ($\mu$mol kg$^{-1}$) | $2312.12 + (10.68 \times T) - (3.50 \times T^2) + (0.16 \times T^3) - (2.42 \times 10^{-3} \times T^4)$ |
| Total alkalinity ($\mu$mol kg$^{-1}$) | $2444.73 - (22.29 \times T) + (0.09 \times T^2) + (1.28 \times 10^{-3} \times T^3) + (4.60 \times 10^{-4} \times T^4)$ |
| Dissolved oxygen ($\mu$mol L$^{-1}$) | $245.85 - (52.01 \times T) + (6.46 \times T^2) - (0.28 \times T^3) + (3.84 \times 10^{-3} \times T^4)$ |
| Nitrate ($\mu$molN L$^{-1}$) | $0.98 \times \dfrac{\left(\dfrac{(DIC \times 35)}{S} - 1977.4\right)}{9.77}$ |
| Ammonium ($\mu$molN L$^{-1}$) | $0.02 \times \dfrac{\left(\dfrac{(DIC \times 35)}{S} - 1977.4\right)}{9.77}$ |
| Phosphate ($\mu$molP L$^{-1}$) | $\dfrac{\left(\dfrac{(DIC \times 35)}{S} - 1961.3\right)}{141.23}$ |

## 3 Results

### 3.1 Overall sea–air CO$_2$ exchange features and comparison with other datasets

Comparison with Surface Ocean CO$_2$ Atlas (SOCAT; Bakker et al., 2016) over the 2010–2019 period shows that the model could capture the high sea surface pCO$_2$ environment within the TMC as shown from underway observation track in the Indonesia Throughflow area (Figure 2). Modeled sea surface pCO$_2$ inside the TMC also fell within the observed value by Kartadikaria et al. (2015) and Hamzah et al. (2020). They reported sea surface pCO$_2$ value inside the TMC that higher than 400 µatm and suggested the role of the region as a net atmospheric CO$_2$ source. However, the model still overestimates the

sea surface pCO$_2$ in open ocean surrounding the TMC such as the Western Pacific Ocean. The striking sea surface pCO$_2$ gradient between open ocean and inner TMC was still reproduced by the model despite being much weaker than SOCAT data. Other dataset derived from empirical model (Iida et al. 2021) and machine learning (Landschützer et al., 2016; not shown) with 1° × 1° horizontal resolution did not exhibit such contrasting differences. Excess alkalinity in the sea surface, calculated as difference between total alkalinity and DIC, from simulation indicates that there is a considerable gradient

between the TMC and surrounding open ocean which corresponds to higher sea surface pCO$_2$ inside the TMC compared with open ocean. Modeled excess alkalinity in the open ocean were still lower compared with reconstruction in Iida et al (2021) which may explain the higher–than–observed sea surface pCO$_2$ in the open ocean.



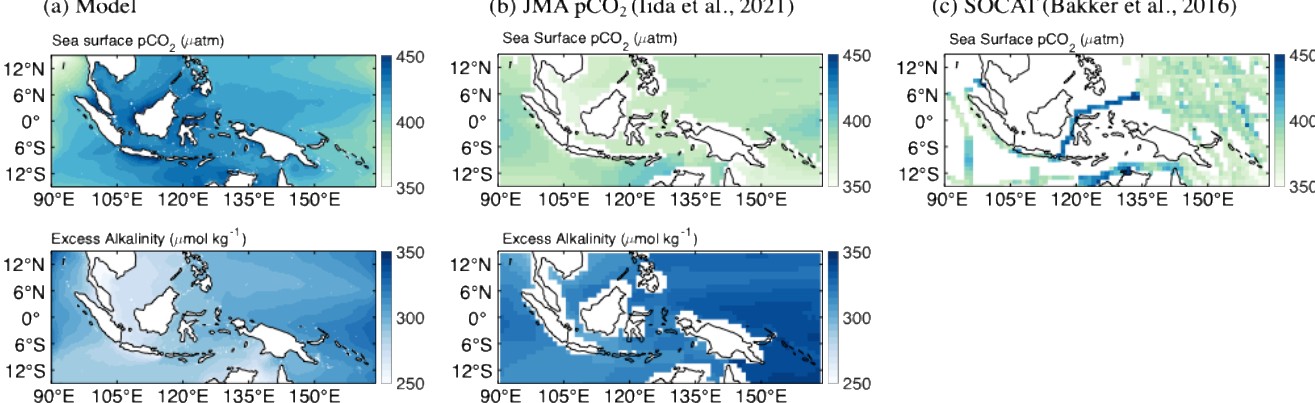

**Figure 2. First row: Overall sea surface pCO₂ (in μatm) indicated by (a) Numerical model utilized in this study, (b) Global reconstruction product using empirical model (Iida et al., 2021), and observation archived in Surface Ocean CO₂ Atlas (SOCAT; Bakker et al., 2016). Second row: Sea surface excess alkalinity (in μmol kg⁻¹) as calculated by (a) Numerical model and (b) empirical model (Iida et al., 2021).**

The atmospheric $CO_2$ sink/source characteristic over the 2010–2019 shows large spread between datasets as shown in Figure 3. Our model indicates that inside TMC acts as a net atmospheric $CO_2$ source with average flux of $+0.42$ mol m$^{-2}$ year$^{-1}$ consistent with study by Kartadikaria et al. (2015) and Hamzah et al. (2020) which estimated the average $CO_2$ degassing rate of $+0.33$ mol m$^{-2}$ year$^{-1}$ and $+0.10$ mol m$^{-2}$ year$^{-1}$, respectively. Overall $CO_2$ source characteristic within the TMC produced by model here also in line with recent state–of–art global scale earth system model which incorporated coastal dynamics (Mathis et al., 2022). Other sea surface pCO₂ and sea–air $CO_2$ exchange reconstruction products in contrast, indicated the area as a net atmospheric $CO_2$ sink with average flux of $–0.04$ mol m$^{-2}$ year$^{-1}$ (Iida et al., 2016) and $–0.01$ mol m$^{-2}$ year$^{-1}$ (Landschützer et al., 2016), respectively. Note that in comparison with study conducted by Kartadikaria et al. (2015), study by Hamzah et al. (2020) utilized much shorter observation data and smaller area.

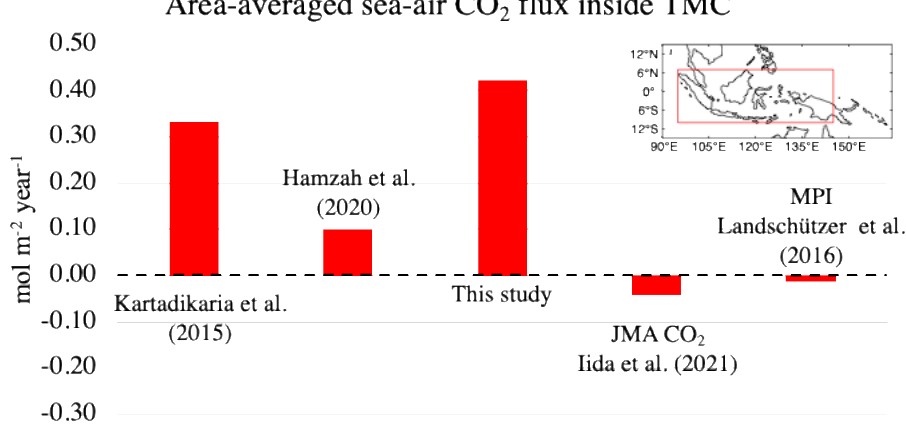

**Figure 3. Estimated sea–air CO₂ flux (in mol m⁻² year⁻¹) inside the TMC according to various studies and global reconstruction product. Positive and negative value indicates atmospheric CO₂ source and sink signature, respectively. Area used for calculating**



the sea–air $CO_2$ flux from simulation experiment and global reconstruction products (i.e, Iida et al., 2021; Landschützer et al., 2016) is bounded by red line shown in upper–right map.

## 3.2 Seasonal pattern of sea–air $CO_2$ exchange

Figure 4 showed mean seasonal cycle of sea surface $pCO_2$ and sea–air $CO_2$ flux across the TMC between 2010–2019. During winter (December–February), model results show high sea surface $pCO_2$ around the Timor Sea along with strong $CO_2$ degassing to the atmosphere. In following spring (March–May), high sea surface $pCO_2$ is more distributed within the TMC from northern part to southern part of the region, including South China Sea and most of area in Indonesia sea. Strong $CO_2$ degassing however, did not apparent during the season as result from weak wind speed. Simulation results indicated strong $CO_2$ degassing appear again during Summer (June–August) particularly around South China Sea and South of Java along with high $pCO_2$ which can exceed 30 gC $m^{-2}$ $year^{-1}$ and 466 µatm, respectively. High sea surface $pCO_2$ and strong $CO_2$ degassing around South of Java was further maintained until Autumn (September–November) making it the longest strong $CO_2$ degassing across the TMC region. The strong $CO_2$ degassing period in South of Java was coincided with upwelling season which previously observed by many studies (Horii et al., 2018; Ningsih et al., 2013: Siswanto et al., 2020; Susanto et al., 2001).



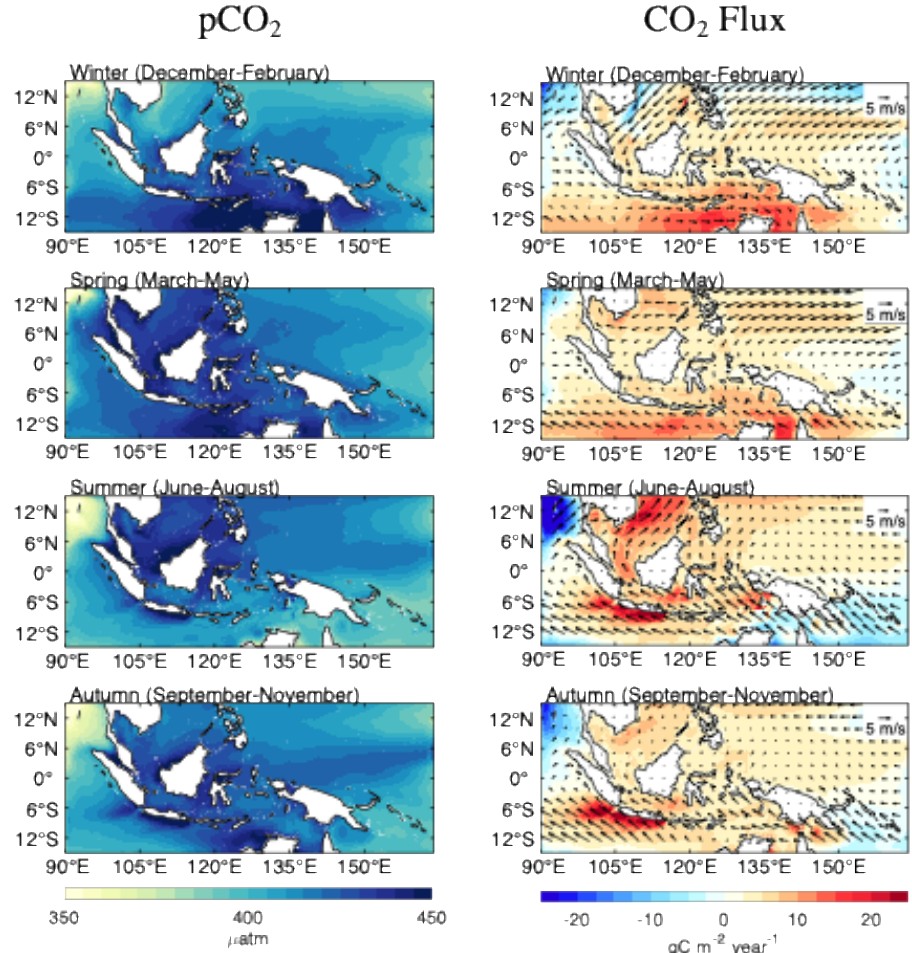

**Figure 4. Mean seasonal cycle of sea surface pCO₂ (Left column figures; in µatm) and sea–air CO₂ flux (Right column figures, in gC m⁻² year⁻¹). Positive and negative value in sea–air CO2 flux map indicates atmospheric CO₂ source and sink area, respectively.**

Figure 5 shows the results of sea surface $pCO_2$ decomposition analysis following Takahashi et al. (1993). It was indicated that spatial variation of sea surface $pCO_2$ closely follows SST seasonality in the TMC. As the region located in the tropics, there is a clear interhemispheric difference of SST seasonal cycle in the northern and southern part of the region which led to alternating north–south gradient of sea surface $pCO_2$. The biological processes, represented by SSDIC and SSAlk, generally has weaker influence than the SST influence on large–scale sea surface $pCO_2$ seasonality as the effect of SSDIC and SSAlk on sea surface $pCO_2$ tends to cancel each other. With the exception on South of Java, the biological process in the region induced net increase(decrease) to the sea surface $pCO_2$ during the onset(termination) of upwelling. In those periods, the biological effect on sea surface $pCO_2$ was counteracted by the SST effect. We also noted that in comparison with the open ocean, our simulation results indicated the inner part of TMC region as biogeochemical activities 'hotspot' where influence



of SSDIC and SSAlk to sea surface pCO$_2$ was more notable. As expected for the sea surface salinity, changes in SSS showed smallest influence on the sea surface pCO$_2$ seasonality across the region compared with other three driving components.

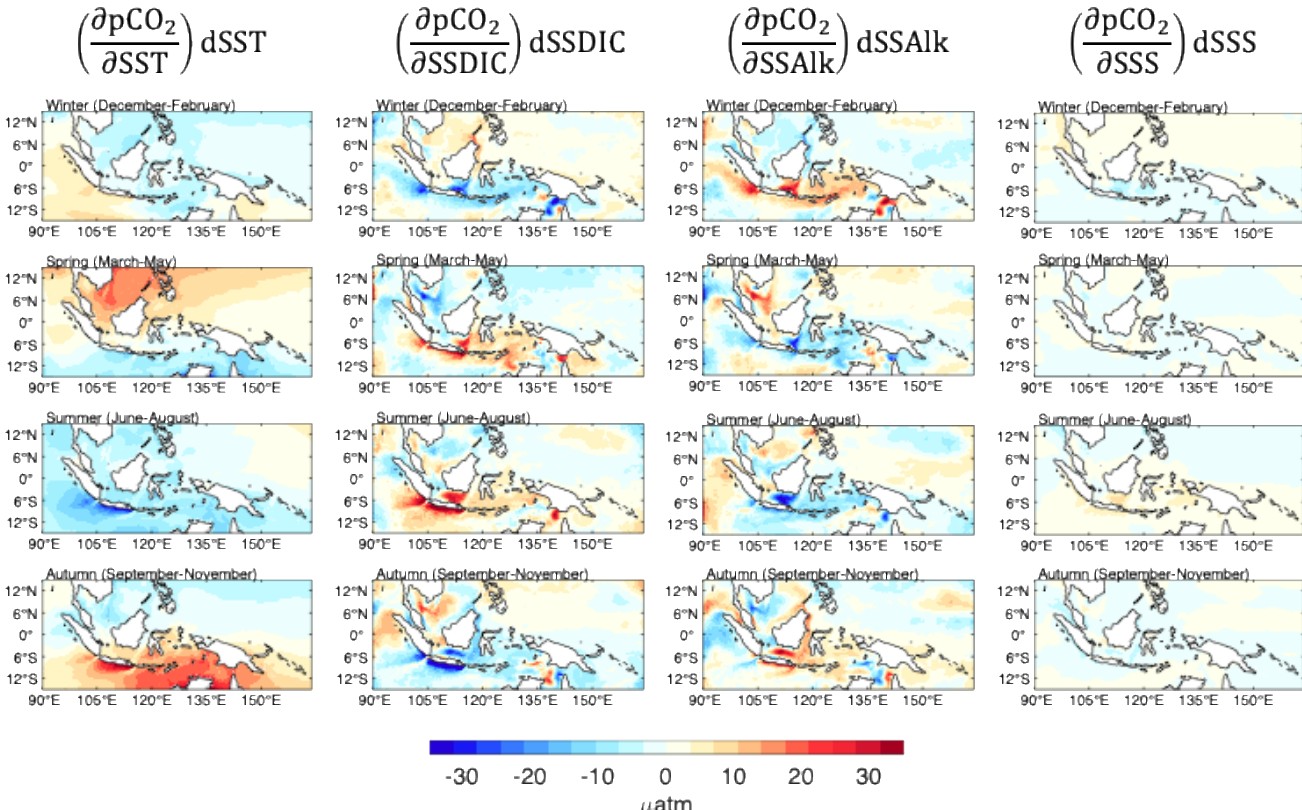

**Figure 5. Decomposition of sea surface pCO$_2$ seasonality across the Tropical Maritime Continent (in μatm). From left column to right column figures: Changes in sea surface pCO$_2$ caused by sea surface temperature, sea surface DIC, sea surface Alkalinity, and sea surface salinity.**

Upwelling in the South of Java reaches its annual peak in September as shown in the sea surface chlorophyll–a concentration from MODIS satellite (Hu et al., 2012) and phytoplankton carbon biomass from simulation results (Figure 6). Calculation of Chl:C ratio in the area shows an average value of 0.02 which within the range in study by Arteaga et al. (2016). The sea surface pCO$_2$ in the South of Java closely follow this upwelling seasonality with sea surface DIC become the main driving component for high sea surface pCO$_2$ during upwelling peak. On the other hand, strong CO$_2$ degassing to the atmosphere (~ 30 gC m$^{-2}$ year$^{-1}$) lead the upwelling peak by one month (August) and maintained until succeeding month of September. Wind speed around the area according to JRA–55 showed its maxima in August which can explain the early strong CO$_2$ degassing prior to the upwelling peak. Although wind speed around south of Java showed relaxation starting from September, surface water with high inorganic carbon concentration further maintains the strong CO$_2$ degassing condition due to high sea surface pCO$_2$.



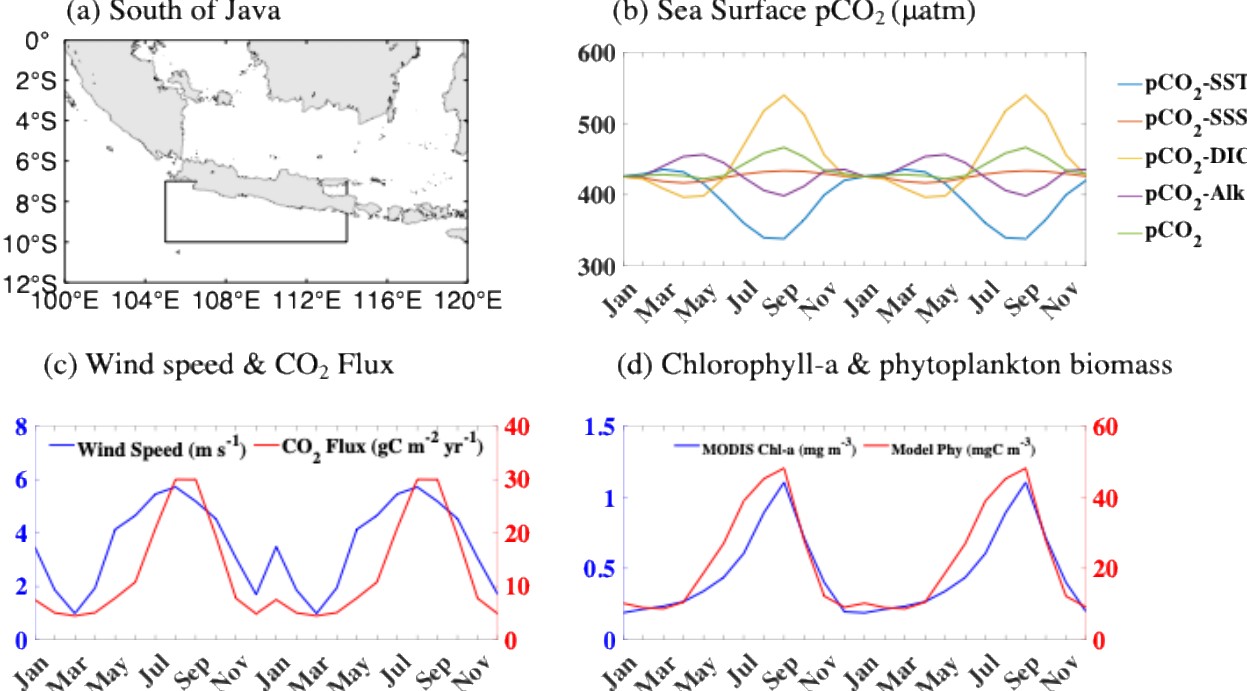

**Figure 6. (a) South of Java map; (b) Modelled seasonal sea surface pCO₂ associated with its component (in μatm); (c) Seasonal wind speed around South of Java according to JRA–55 datasets (in m/s) and modelled sea–air CO₂ flux (in gC m⁻² year⁻¹); (d) Modelled seasonal phytoplankton carbon biomass (mgC m⁻³) and chlorophyll–a concentration (mg m⁻³) according to Moderate Resolution Imaging Spectrometer (MODIS) satellite observation**

### 3.3 Interannual variability over the last decade (2010–2019)

The interannual variability was examined by removing the mean seasonal cycle of simulated sea surface pCO₂ and sea–air CO₂ flux over the 2010–2019 period which later identified as anomalies. In addition to the secular trend caused by atmospheric CO₂ concentration growth used in the model forcing, modeled sea–air CO₂ flux showed notable interannual variation particularly between 2015–2016 and 2019 coincided with the development of 2015/2016 El Niño and 2019 pIOD (Figure 7). Those two major modulations in the sea–air CO₂ flux had shown to slow down the decreasing trend in the CO₂ flux over the study period (Figure 7a). The sea surface pCO₂ anomalies on the other hand, showed relative steady increase although some modulations like in the early 2016 can be observed. This again confirms that sea–air CO₂ exchange across the TMC also subject to low–frequency modulation which can be related to the Indo–Pacific climatic forcing.





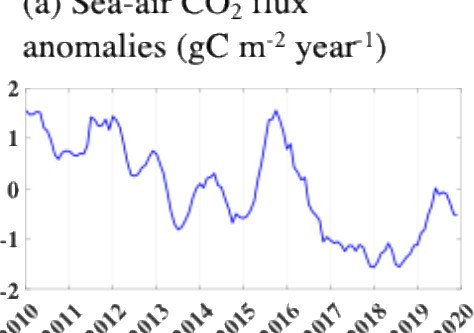
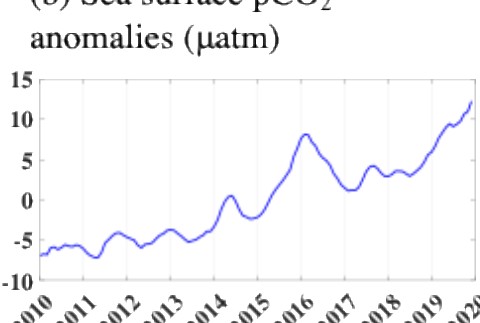

**Figure 7. Five–months moving average of (a) sea–air CO₂ flux anomalies (in gC m⁻² year⁻¹) and (b) sea surface pCO₂ anomalies (in µatm) across the 95ºE–164ºE; 15ºS–15ºN over the 2010–2019 period. Positive and negative values indicate higher–than–usual sea surface sea–air CO₂ flux/ pCO₂ and lower–than–usual relative to the mean seasonal cycle, respectively.**

Quantification of Indo–Pacific climatic forcing influence on the sea–air $CO_2$ exchange across the TMC was conducted by regression analysis on sea surface $pCO_2$ anomalies, sea–air $CO_2$ flux anomalies, and wind speed anomalies. Considering that ENSO and IOD showed a statistically significant correlation over the 2010–2019 (Pujiana et al., 2019), we further separated the effect of ENSO on IOD and *vice versa* by performing partial correlation analysis for sea surface $pCO_2$ anomalies, sea–air $CO_2$ flux anomalies, and wind anomalies following the methods in Saji and Yamagata (2003) prior to the regression analysis. This will allow us to evaluate the extent of each climatic forcing effects on sea–air $CO_2$ exchange across the study region. The ENSO and IOD events were represented by central–eastern Pacific sea surface temperature anomaly 'NINO3.4' and dipole mode index 'DMI' (See Saji et al., 1999 for further details about DMI calculation), respectively from HadISST 1.1 (Rayner et al. 2003). For uniformity reasoning, we regressed the sea surface $pCO_2$ anomalies, sea–air $CO_2$ flux anomalies, and wind speed anomalies from JRA55 against one–standard deviation ($\pm 1\sigma$) of NINO3.4 ($\sigma_{NINO3.4} = 0.80$ ºC) and DMI ($\sigma_{DMI} = 0.26$ ºC) over the 2010–2019 period. Note that typical ENSO mature phase occurs within the November–March period, while the IOD occurs in July–November. Thus, results presented in this section will be focused on these two–period which corresponds to each climatic event.

Regressed sea surface $pCO_2$ anomalies and sea–air $CO_2$ flux anomalies against one–standard deviation ($\pm 1\sigma$) of NINO3.4 and DMI revealed distinguishable spatial extents of modulation (Figures 8 and 9). Anomalies associated with ENSO during the November–March period tended to have a larger spatial extent, extended from SETIO region up to South China Sea, compared with IOD during July–November, which was confined along south of Java. Results from the regression analysis also showed an extended minor influence of IOD on the sea surface $pCO_2$ variabilities up to the lesser Sunda Island water area and inside the Indonesian seas.

Regressed sea surface $pCO_2$ anomalies against NINO3.4 further suggested a stronger sensitivity of the TMC to ENSO forcing compared with the adjacent Western Pacific Ocean denoted by the area which satisfy statistical significance threshold defined in this study (i.e., $p < 0.01$). The smaller extent of regressed sea–air $CO_2$ flux anomalies against NINO3.4 implies the nonlinearity between sea surface $pCO_2$ modulation and sea–air $CO_2$ exchange modulation in response to the same



climatic forcing which is possible considering the additional factor of wind speed that determine the sea–air $CO_2$ flux.

300 Regression analysis conducted here suggested an anomalous sea surface $pCO_2$ and $CO_2$ degassing increase associated with associated with ENSO over the 2010–2019 can be as high as +40 µatm $(1\sigma NINO3.4)^{-1}$ and +5 gC m$^{-2}$ year$^{-1}(1\sigma NINO3.4)^{-1}$. Exclusion of the 2015/2016 El Niño event in the regression analysis (Figure 8, second column) decreased the magnitude of the sea surface $pCO_2$ anomalies and significantly reduced the spatial extent of the sea–air $CO_2$ flux anomalies despite the occurrence of strong 2010–2012 La Niña. This further suggests that the double–dip La Niña in 2010/2011 and 2011/2012

305 induced less–pronounced sea–air $CO_2$ flux modulation within the TMC.

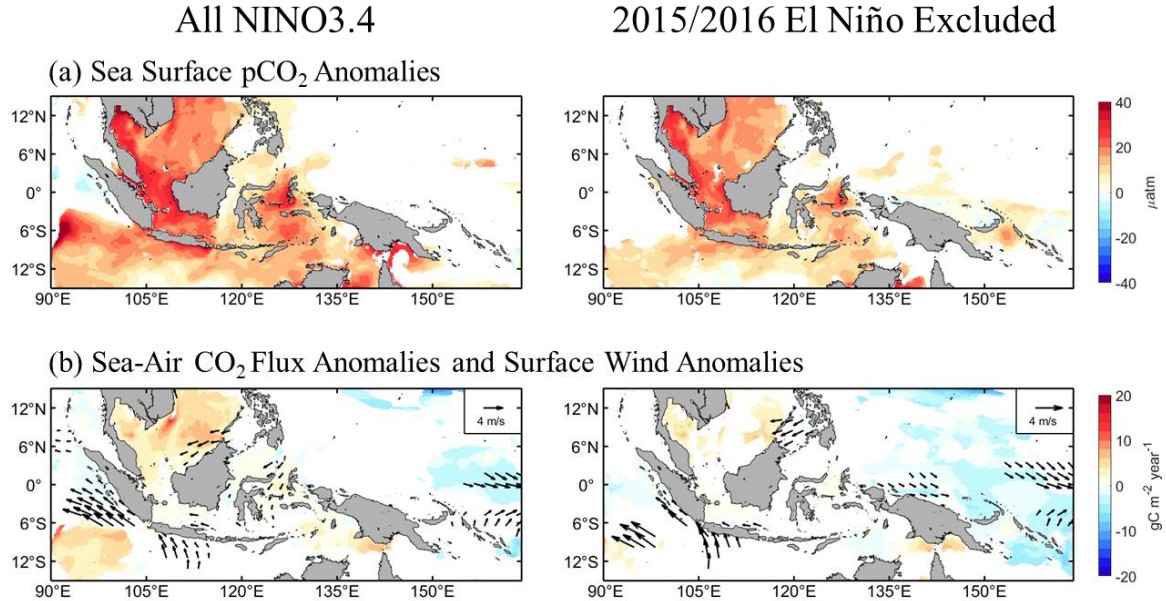

**Figure 8. November–March (a) regressed sea surface pCO₂ anomalies (in µatm) and (b) regressed sea–air CO₂ flux anomalies (in gC m⁻² year⁻¹) along with wind anomalies (Vector arrows, in m/s) against one–standard deviation of NINO3.4 (1σNINO3.4) at zero–lag. Regression was calculated after separating IOD effect on ENSO. Shaded colours and vector arrows are significant at *p* <**
310 **0.01.**

Despite regressed sea surface $pCO_2$ anomalies against DMI, which showed a magnitude comparable to the NINO3.4– regressed value which about +40 µatm $(1\sigma DMI)^{-1}$, the regressed sea–air $CO_2$ flux anomalies against DMI showed a much higher value. Regressed sea–air $CO_2$ flux anomalies against the DMI showed value as high as +20 gC m$^{-2}$ year$^{-1}(1\sigma DMI)^{-1}$. Removing the 2019 pIOD events from the regression analysis (Figure 9, second column) resulted in only slight changes in

315 both sea surface $pCO_2$ anomalies and sea–air $CO_2$ flux anomalies associated with IOD. This implies that even a typical IOD event (after the ENSO influence has been removed) could trigger strong anomalies in the sea–air $CO_2$ flux, especially around the south of Java.





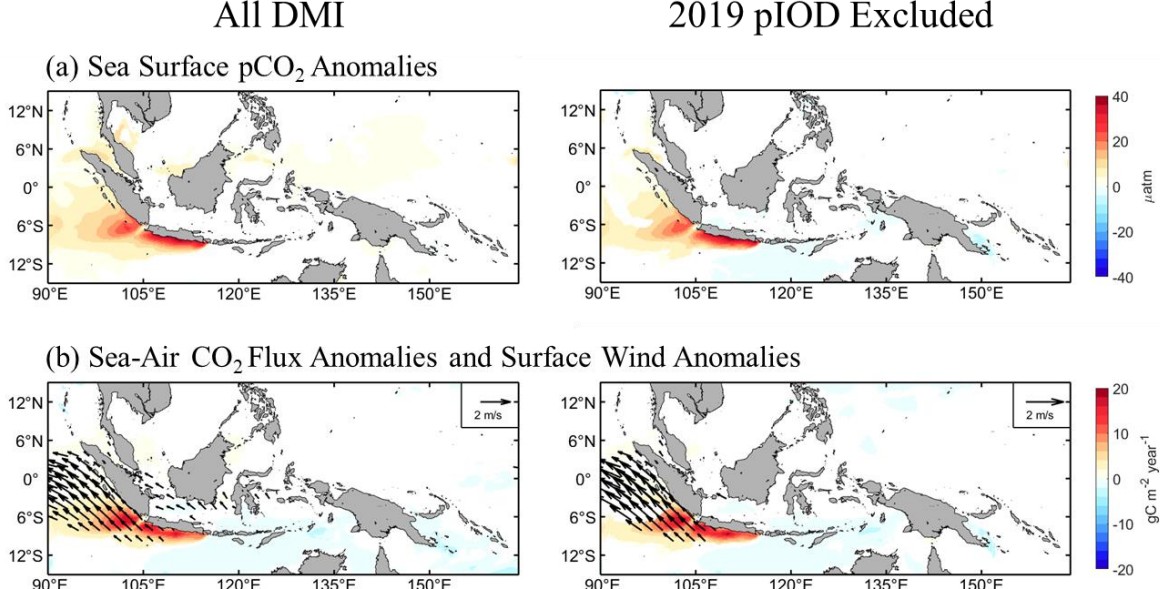

**Figure 9. July–November (a) regressed sea surface pCO₂ anomalies (in μatm) and (b) regressed CO₂ flux anomalies (in gC m⁻².**
**year⁻¹) along with wind anomalies (Vector, in m/s) against one–standard deviation of DMI (1σDMI) at zero–lag. Regression was**
**calculated after separating ENSO effect on IOD. Shaded colours and vector arrows are significant at $p < 0.01$.**

## 4 Discussion

Examination of long–term sea–air $CO_2$ exchange in the TMC requires careful consideration. Global sea surface $pCO_2$ and
sea–air $CO_2$ flux reconstruction based on neural network approach (Landschützer et al., 2016) or empirical model (Iida et al.,
2021) shows opposite atmospheric $CO_2$ sink/source characteristic with observations–based study (Hamzah et al., 2020;
Kartadikaria et al., 2015). Lack of continuous observation system (e.g., open ocean mooring) in the area create additional
constraint to decide the true atmospheric $CO_2$ sink/source characteristic of the region. Our coupled OGCM simulation
experiment on the other hand, showed consistent results with those observations–based study within the TMC and other
model with higher complexity. Consistency in the $CO_2$ sink/source characteristic shown by our simulation experiment
enabled us to further analyse the produced sea–air $CO_2$ exchange variabilities in response to the climatic forcing over the
2010–2019 period. Biases in open ocean as shown previously can be associated to the excess alkalinity in the surface that
still underestimated in the model. This was supported by the fact that the model domain is still within the Indo–Pacific warm
pool region with small horizontal SST gradient which should results in similar thermally forced sea surface $pCO_2$. Further
model improvement to address this issue by enhancing the excess alkalinity gradient between TMC and its adjacent open
ocean is subject to upcoming studies.

While acted as full–year net atmospheric $CO_2$ source, particularly inside the TMC, our simulation results indicated
pronounced seasonality around the South of Java. Strong seasonal winds that triggered upwelling around the area during





autumn (Horii et al., 2018; Ningsih et al., 2013: Siswanto et al., 2020; Susanto et al., 2001) created favourable conditions for strong $CO_2$ degassing through a combination of accelerated gas exchange and an abundant supply of subsurface inorganic

carbon. This mechanism was not apparent in other areas across the TMC, making it a unique feature from the sea–air $CO_2$ exchange perspective. Note that the aggregate results of upwelling to the sea–air $CO_2$ exchange might vary across regions (Chakraborty et al., 2018; Valsala et al., 2014) and thus, the results presented by this model for the South of Java should not be taken as a generalization for all upwelling–active regions.

An attempt to elucidate the extent of extreme climate events (2015/2016 El Niño and 2019 pIOD) influence on the sea–air

$CO_2$ exchange across the TMC through regression analysis yielded notable results. In agreement with the suggestion by Kartadikaria et al. (2015), regression analysis suggested a lower–than–usual sea surface $pCO_2$ during La Niña and *vice versa* during El Niño. The sea–air $CO_2$ flux modulation under ENSO influence interestingly, did not necessarily follow the sea surface $pCO_2$ modulation spatial pattern and relatively weaker than under IOD influence. This can be attributed to different modulation tendency caused by the two climatic forcings.

Weaker (stronger) northwest monsoon circulation within the TMC during El Niño (La Niña) due to anomalous divergence (convergence) in the western Pacific could weaken (strengthen) the gas exchange between the sea surface and the atmosphere. However, shifts in the Walker circulation (Alexander et al., 2002) caused by the same anomalous divergence (convergence) also altered the cloud distribution across the tropics, including the TMC itself, and affected SST. Decreased (increased) cloud cover around the TMC during El Niño (La Niña) can increase (decrease) SST through an increase

(decrease) in incoming solar radiation. This mechanism could increase (decrease) sea surface $pCO_2$ and ultimately strengthen (weaken) $CO_2$ degassing. The opposite modulation tendencies between atmospheric and oceanic conditions in response to ENSO forcing made the $CO_2$ flux anomalies magnitude associated with ENSO less pronounced, despite the strong sea surface $pCO_2$ anomalies.

Conversely, the IOD did not exhibit such opposite tendencies, which resulted in the strong linearity between the sea surface

$pCO_2$ anomalies and the sea–air $CO_2$ flux anomalies. Typical IOD events occur between late summer–autumn, where seasonal upwelling occurs (Delman et al., 2016; 2018; Susanto et al., 2001). Anomalous south–easterly (north–westerly) winds during the pIOD (nIOD) around SETIO can directly modulate upwelling around the south of Java. Enhanced (suppressed) upwelling in response to stronger (weaker) wind forcing during pIOD (nIOD) then result in higher (lower)–than–usual sea surface $pCO_2$ from the ocean side and accelerated (decelerated) gas exchange on the atmospheric side.

Exclusion of extreme climate events (i.e., 2015/16 El Nino and 2019 pIOD) in the regression analysis implies the events were likely responsible for a larger extent of sea–air $CO_2$ exchange modulation around the TMC over the last decade. Significant difference in regressed sea–air $CO_2$ flux anomalies after exclusion of 2015/16 El Niño despite the existence of strong 2010–2012 La Niña emphasizes the peculiarities of the recent extreme El Niño event. The 2010–2012 La Niña shows comparable magnitude with the 2015/16 El Niño as indicated by the Multivariate ENSO Index v2 (MEIv2; Zhang et al.,

2019). It is possible that the Pacific decadal climatic shift in the 2010s modified the ENSO flavour, as pointed out by Newman et al. (2016), including its influence on the TMC, so that sea–air $CO_2$ flux modulation related to the 2015/2016 El



Niño showed substantially different characteristics. Further modelling studies across the TMC over a longer time–scale will be needed to confirm this possible decadal variation in the sea–air $CO_2$ exchange. Considering the simulation results and analysis conducted here, pronounced modulation of the sea–air $CO_2$ exchange across the TMC in the future can be expected,

as recent studies have indicated an intensification of extreme climate anomalies under the effect of greenhouse gas forcing (Cai et al., 2018; Grothe et al., 2019; Zhang et al., 2018).

Finally, one of the biggest challenges hindering this study was that we have not incorporate river discharge in the simulation experiment. The lack of reliable datasets, especially for carbonate chemistry–related parameters, such as total DIC, total alkalinity, and nutrients, as highlighted by Valsala et al. (2014), was the main reason for this limitation. Such data are critical

for evaluating the robustness of any regional–scale watershed modelling effort before further use in coupled OGCM–ecosystem models. Incorporating river discharge inappropriately for studying upper–ocean carbon cycle variability will only produce questionable results. DIC concentration from river discharge, for example, varies widely between river mouths, with values ranging from 284 µmol kg$^{-1}$ (Rosentreter and Eyre, 2019) to as high as 3,500 µmol kg$^{-1}$ (Kawahata et al., 2000). This highly variable value did not include the possible strong seasonal and interannual variability of the river–discharged

material, as presumed by Xiu and Chai (2014).

## 5 Conclusion

This study presents results from high–resolution coupled OGCM with low–trophic ecosystem simulation experiment focusing on sea–air $CO_2$ exchange variabilities across the TMC, a region regarded as undersampled (Hamzah et al., 2020) and usually overlooked by global–scale modelling and/or reconstruction efforts. Compared with available reconstruction

product, simulated atmospheric $CO_2$ sink/source characteristic within the TMC from this modelling study agreed with previous observation–based studies (Hamzah et al., 2020; Kartadikaria et al., 2015) where the region acts as net atmospheric $CO_2$ source. Further, we also analysed the interannual variations of sea–air $CO_2$ exchange under Indo–Pacific climatic forcing over the 2010–2019 period along with its possible mechanism from sea–air interaction perspective. This has never been done before considering the aforementioned limitations. Generally, the $CO_2$ degassing anomalies showed in–phase

relationship with both IOD and ENSO (i.e., positive sea–air $CO_2$ flux anomalies during positive phase of IOD or ENSO, and *vice versa*). The ENSO tends to induce larger scale of sea–air $CO_2$ flux modulation while the IOD showed confined influence around South of Java but with higher magnitude of modulation. It was further suggested that the latest extreme climate event such as the 2015/2016 El Niño and 2019 pIOD were responsible for slowing the secular trend in the $CO_2$ degassing to atmosphere from the region. While results presented may provide insight about sea-air $CO_2$ exchange

variabilities in TMC, it can be utilized also to invite interdisciplinary research collaborations regarding establishment of a continuous sea–air $CO_2$ exchange monitoring system across the region and enrich our understanding of its dynamics under changing environments.

**Code Availability**

The original code of our model and necessary code to prepare the input for simulation experiment can be accessed at
https://github.com/NakamuraTakashi.

**Data Availability**

Authors declare that all the data used in this study area publicly available. The Surface Ocean $CO_2$ Atlas (SOCAT) datasets can be accessed through https://www.socat.info. Global sea surface temperature from HadISST1.1 used for calculating the NINO3.4 and DMI can be retrieved from https://www.metoffice.gov.uk/hadobs/hadisst/. Global sea surface $pCO_2$ and sea–
air $CO_2$ flux reconstruction from Japan Meteorological Agency (JMA) and can be retrieved from https://www.data.jma.go.jp/gmd/kaiyou/english/co2_flux/co2_flux_data_en.html. Global sea surface $pCO_2$ and sea–air $CO_2$ flux using two–step neural network produced by the Max Planck Institute can be obtained from https://www.nodc.noaa.gov/archive/arc0105/0160558/3.3/data/0–data/

**Author Contributions**

FA and TN designed the study and conduct the numerical experiment. AW and ARK provided $pCO_2$ observation data within the study area and help interpret the results. KN acquired the funding. FA wrote the original draft of the manuscript. All authors have read and give their review and comment to improve the final manuscript.

**Competing Interest**

The authors declare that they have no conflict of interest

**Disclaimer**

Publisher's note: Copernicus Publications remains neutral with regard to jurisdictional claims in published maps and institutional affiliations.

**Acknowledgement**

This study was funded by Science and Technology Research Partnership for Sustainable Development through the
"Comprehensive Assessment and Conservation of Blue Carbon Ecosystem and their Services in the Coral Triangle (BlueCARES)" Project. The first author is a scholarship recipient from Japan's ministry of education, culture, sports, science,





and technology (MEXT). We are grateful for the high–performance computing services 'TSUBAME 3.0' provided by the Tokyo Institute of Technology for making this simulation experiment possible.

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
