# Peer review of "Seasonal to interannual variabilities of sea—air CO2 exchange across Tropical Maritime Continent indicated by eddy—permitting coupled OGCM experiment"

_EGUsphere, 2022_

## Referee Comment (RC1)

Review of "Seasonal to interannual variabilities of sea–air CO2 exchange across Tropical Maritime Continent indicated by eddy–permitting coupled OGCM experiment" by Amri et al submitted to Ocean Science

The paper is a long-standing pending issue in addressing the CO2 fluxes of this very important region. It deserves publication in Ocean Science. However, it has missed the following key elements, which need to be addressed.

Valsala et al., (2020) have done, for the first time, analysis of 60-year long record of sotheastern tropical Indian Ocean CO2 flux variability, pCO2 variability associated with the IOD. The study concluded that, "*The IOD leads to a substantial sea-to-air $CO_2$ flux variability in the southeastern tropical Indian Ocean over a broad region (70–105°E, 0–20°S), with more focus near the coast of Java-Sumatra due to the prevailing upwelling dynamics and associated westward propagating anomalies. The sea-to-air $CO_2$ fluxes, surface ocean partial pressure of $CO_2$ (pCO2), the concentration of dissolved inorganic carbon (DIC), and ocean alkalinity (ALK) range as much as $\pm1.0$ mole $m^{-2}$ $year^{-1}$, $\pm20$ μatm, $\pm35$ μmole $kg^{-1}$, and $\pm22$ μmole $kg^{-1}$ within 80–105°E, 0–10°S due to IOD. The DIC and ALK are significant drivers of pCO2 variability associated with IOD. The roles of temperature (T) and biology are found negligible. A relatively warm T and extremely high freshwater forcing make the southeastern tropical Indian Ocean carbon cycle variability submissive to DIC and ALK evolutions in contrast to the tropical eastern Pacific where changes in DIC and T dominate the pCO2 interannual variability. For the first time, this study provides a most comprehensive and extended analysis for the region while highlighting significant differences in carbon cycle variability of the eastern tropical Indian Ocean compared to that of the other parts of the global oceans.*"

This is an important recent work and needs to be cross-discussed in this paper, especially due to the reason the IOD impacts are revisited in this manuscript, and the results have differences. It is always good to have various modelling comparisons so that the community is benefited from knowing how the model performs and differ from each other. Other papers missed addressing are also added below:

Valsala, V., M. G. Sreeush, and K. Chakraborty, (2020), IOD impacts on Indian the Ocean Carbon Cycle, Journal of Geophysical Research, https://doi.org/10.1029/2020JC016485

Chakraborty K., V. Valsala, T. Bhattacharya, J. Ghosh, (2021), Seasonal cycle of surface ocean pCO2 and pH in the northern Indian Ocean and their controlling factors, Progress in Oceanography, Vol.198, doi.org/10.1016/j.pocean.2021.102683

Valsala V., Sreeush M.G., Anju M., Sreenivas P., Tiwari Y.K., Chakraborty K., Sijikumar S., An observing system simulation experiment for Indian Ocean surface pCO2 measurements,

Progress in Oceanography, 194: 102570, June 2021, DOI: 10.1016/j.pocean.2021.102570, 1-14

Sreeush, M. G., Valsala, V., Pentakota, S., Prasad, K. V. S. R., and Murtugudde, R (2018), Biological production in the Indian Ocean upwelling zones – Part 1: refined estimation via the use of a variable compensation depth in ocean carbon models, Biogeosciences, 15, 1895-1918, https://doi.org/10.5194/bg-15-1895-2018

---

## Author Comment (AC1)

Response to Referee #2 Comment (RC2)

We would like to thank the referee for taking the time to review our manuscript and for providing constructive comments. Please see our responses to the comments below.

Amri et al. examined $pCO_2$ and air-sea $CO_2$ fluxes variability over the Tropical Maritime Continent (TMC) using a regional ocean biogeochemical (BGC) model. Surface $pCO_2$ patterns across the TMC have not been well constrained, so this study represents a valuable effort to better understand carbon system dynamics in the region. However, I have three major concerns about the model results and analysis:

**Response**: Thank you for your comments on our manuscript. We have addressed the comments suggested by the referee as follows:

**1)** It is not clear to me whether the model is getting realistic $pCO_2$ patterns or not. The comparison with Bakker et al. (2016), Iida et al. (2022), and Landschützer et al. (2016) suggests a significant overestimation of surface $pCO_2$, especially in the open ocean region. I wonder to what degree the initial and boundary conditions for the BGC model, derived through an analytical (regression models) approach, were properly resolved. Since the authors do not provide a model validation, neither physic or biogeochemistry –putting aside $pCO_2$ and $CO_2$ fluxes–, it is difficult be confident in their results. I think this study requires a proper model validation, which should include model-data comparisons for horizontal and vertical patterns of temperature, salinity, nutrients, and carbon system variables when available.

**Response**: First, we would like to inform the referee that we have performed another simulation experiment with an identical configuration as in the submitted manuscript but with a longer period and different atmospheric $CO_2$ concentration scenarios. The simulation was conducted from January 1994 to December 2020 under two atmospheric $CO_2$ scenarios. The first scenario used a "controlled" atmospheric $CO_2$ concentration at the monthly level in 1994 during the model integration period (Named CTL scenario). Another scenario employed "realistic" atmospheric $CO_2$ concentration, where it follows the global monthly average concentration from January 1994 to December 2020 (Named HIS scenario), as recorded in the Earth System Research Laboratory of the National Oceanic and Atmospheric Administration (ESRL NOAA). We confirmed that the HIS scenario produced a similar overall $pCO_2$ and $CO_2$

flux trend, as in the submitted manuscript. By performing longer simulation, there is a higher chance for us to utilize observation data that coincide with the simulation period.

The results from this experiment were compared with observation datasets surrounding the study area, such as the tropical Pacific mooring array (i.e., TAO-TRITON), high-resolution climatological reconstruction products from WOA 2018, and observation data archived in GLODAP v2 (Key et al., 2004). Similar to the submitted manuscript, we considered the first two years of simulation as the spin-up period; therefore, the analyzed results were based on the 1996-2020 period. Overall, we are delighted to inform the referee that our model shows promising results with these observations, implying that the model configuration used here was robust enough to approach the physical processes in the area. Please check the comparison results provided in Figures AR1–AR4. We will include these results in the revised manuscript.

[Figure]

(a) Mooring locations

(b) Time-series of sea surface temperature (in °C)

[Figure]

(c) Time-series of sea surface salinity (in psu)

**Figure AR1. (a) Mooring locations for modeled sea surface temperature and salinity validations from 1996 to 2020; time series of monthly (b) sea surface temperature (in °C) and (c) sea surface salinity (in psu) according to simulation results (blue solid line) and observations (red solid line) during 1996-2020 period.**

(a) Mooring locations

[Figure]

(b) Time-series of water velocity at 10m depth

[Figure]

**Figure AR2. (a) Mooring locations for modeled water velocity validations and (b) time series of monthly water velocity (zonal and meridional component; in cm s⁻¹) according to simulation results (blue solid line) and observations (red solid line).**

The horizontal distribution for both sea surface temperature and salinity was also consistent with recent World Ocean Atlas 2018 (WOA18) climatological mean which recently has relatively high horizontal resolution ($0.25^o \times 0.25^o$) compared with other parameters, especially the biogeochemical fields, which still in coarse resolution ($1^o \times 1^o$ to $5^o \times 5^o$). The overall pattern correlation between simulation and WOA18 for sea surface salinity ranged from 0.83 to 0.87. Similar pattern consistency was also achieved for sea surface temperature, where the overall pattern correlation ranged between 0.76 and 0.91.

[Figure]

**Figure AR3. Horizontal distribution of mean seasonal (a) sea surface temperature and (b) sea surface salinity according to simulation results (First and third column figures) and World Ocean Atlas 2018 (Second and forth column figures)**

As for the biogeochemistry parameter, we chose to compare the total dissolved inorganic carbon (DIC) and total alkalinity (TA) from simulation results with observed data archived in GLODAPv2 (Key et al. 2004). We also compared the simulated vertical profiles for both temperature and salinity, as observed in the GLODAPv2 datasets. The comparison divided into two main sections comprising equatorial Pacific Ocean section and equatorial Indian Ocean section which closest to the area of interest (i.e., Tropical Maritime Continent) as shown in Figure AR4. Overall, the model also captured the observed vertical profiles for both the physical (Temperature and Salinity) and biogeochemical parameters (DIC and TA). We acknowledge that there were still notable differences between the simulated and observed TA around the Equatorial Pacific, which created room for improvement in future studies. We decided to use the actual archived observation instead of the gridded product of GLODAPv2 because of the coarse horizontal resolution (i.e., $1° \times 1°$) and inhomogeneous spatiotemporal in-situ measurements surrounding the Tropical Maritime Continent. A study by Lee et al. (2019) raised the question of the gridded product of parameters obtained in an inhomogeneous spatiotemporal sampling manner and further going through smoothing during its gridding processes.

[Figure]

(a) GLODAPv2 sampling points

**Figure AR4. Vertical profile distributions of water temperature, salinity, total dissolved inorganic carbon, and total alkalinity according to the simulation results and observations archived in GLODAPv2. Vertical profile in (b) and (c) were based on HIS simulation scenario**

**2)** The authors claim that changes in sDIC and sAlk represent biological processes, which is not correct. DIC and alkalinity also can change due to advection and mixing, and air-sea flux in the case of DIC. This wrong assumption led to a wrong interpretation for the Taylor decomposition analysis. The authors need to revise that interpretation, making clear that process like wind-driven upwelling of DIC-rich subsurface water could play an important role in the $pCO_2$ variability off Java.

**Response**: Thank you for pointing out this matter, we will revise the interpretation in explaining sea-air $CO_2$ exchange dynamics especially for the South of Java region. Indeed, one

of the key points of high $pCO_2$ in the south of Java is upwelled DIC-rich subsurface water driven by wind. Strong wind speed in the area during the upwelling season further creates favorable conditions for $CO_2$ degassing and makes the area one of the strongest $CO_2$ degassing areas, according to the simulation results.

**3)** The analysis of the interannual $pCO_2$ variability is interesting, but the link to ENSO and IOD need a better explanation. If the patterns are properly described, this could be the most interesting part of the study. Please provide a better description. One thing that catch my attention was the negative trend in the $CO_2$ flux. The authors did not offer any explanation for this trend. I wonder whether this a model $pCO_2$ drift or not.

**Response**: As we have conducted long-term simulation experiments with two atmospheric $CO_2$ scenarios, we concluded that the positive trend in $pCO_2$ and negative trend in the $CO_2$ flux are both the model's response to the increasing atmospheric $CO_2$ concentration, as shown in the figure below. The negative trend in the $CO_2$ flux was due to the growth of atmospheric $CO_2$, which far outpaced the oceanic response. This twin experiment suggested that for areas that act as atmospheric $CO_2$ sources, such as the Tropical Maritime Continent, rapid growth in atmospheric $CO_2$ causes the difference between $pCO_2$ and atmospheric $CO_2$ to become smaller, resulting in a negative trend in the $CO_2$ flux.

[Figure]

**Figure AR5. Time-series of (a) atmospheric CO₂ concentration (in ppm) used in CTL (Red solid line) and HIS (Blue solid line) simulation scenario; (b) Area-averaged pCO₂ anomalies (in μatm); (c) Area-averaged anomalous differences between pCO₂ and atmospheric CO₂ (δpCO₂; in μatm); and (d) Area-averaged sea-air CO₂ flux anomalies (in gC m⁻² year⁻¹)**

We also would like to note that the overall regressed CO₂ flux anomalies and pCO₂ against both NINO3.4 and DMI are still the same using a longer experiment, as shown in Figure AR6. The overall key points related to its variability in response to the Indo-Pacific climatic forcing were not only maintained from the initial submitted manuscript, but also will be enriched since we now have simulation results with unperturbed atmospheric CO₂ conditions.

We will make some revisions to the discussion about the variability patterns associated with the Indo-Pacific climatic forcing (ENSO and IOD). We offer additional figures to explain the possible mechanistic relationship between ENSO and the sea-air CO₂ flux modulation in the last part of the specific comment section. We hope that the additional figures will help readers to understand the proposed mechanism on how climate variability modulates the sea-air CO₂ exchange in the TMC. The mechanism linking ENSO was highlighted because our simulation results suggested a counteracting effect between the atmosphere and the ocean side, which made the CO₂ flux modulation magnitude smaller than that under IOD influence.

[Figure]

**Figure AR6. Regressed (a) pCO₂ anomalies (in μatm) and (b) sea-air CO₂ flux anomalies (in gC m⁻² year⁻¹) against one standard deviation of NINO3.4 SSTA (left figures) and DMI (right figures). All figures are based on the HIS simulation scenario from 1996-2020 period. Shaded color and vector arrows were significant at $p > 0.01$**

Specific comments.

69: I would rather use the name "regional ocean biogeochemical model" instead of OGCM.
**Response**: We will consider your suggestion while also taking into account the Ocean Science policy and whether it is possible to change the title at this stage.

90: I would indicate that "coccolithophores decrease alkalinity, as they produce a body shelf structure made of CaCO3"
**Response**: Thank you for the correction, we will revise the sentence in Line 90.

122-125: I do not understand why you are indicating this Taylor series decomposition here. Need to explain the motivation.
**Response**: The motivation was to explain how we obtained the results presented in Figure 5 in the submitted manuscript. We apologize for not being clear about the motivation.

157: I wonder how your estimated fields for the biogeochemical (BGC) variables compare with the WOA2019 (NO3, PO4, O2). Also, I wonder if you made any comparison between your BGC estimates and BGC fields from reanalysis products (e.g., GLORYS Mercator Ocean).
**Response**: We believe the referee refers to WOA2018 because we could not find any information about WOA2019. We checked the data and found that the horizontal resolution of the data was $1^o \times 1^o$. As mentioned previously, there are at least two issues in utilizing low-resolution gridded products for field comparisons.

This is our first time hearing the GLORYS Mercator Ocean product, although the data record was relatively limited for the biogeochemistry reanalysis product (May 2019–present), it still provides useful data for conducting intermodel comparison. We will also compare our modeled PO4 and NO3 with the same GLODAPv2. Thank you for introducing us to the GLORYS Mercator Ocean product.

Table 2. Alkalinity usually co-varies with salinity. I wonder why you left alkalinity as a function of temperature instead of salinity.
**Response**: We first confirmed the co-variability between salinity/temperature and marine inorganic carbon parameters (DIC and TA) in the modeling domain from observation data archived in GLODAPv2 using the coefficient of determination ($r^2$). The observed DIC and TA in GLODAPv2 showed a higher $r^2$ with water temperature from the same GLODAPv2 record

(See Figure AR7 below). Thus, we proceeded to analytically estimate the DIC and TA for both initial and boundary condition in the model as function of temperature.

[Figure]

**Figure AR7. Comparison of scatter plot between Salinity-TA (Left figure) and Temperature-TA (Right figure)**

188: It would be helpful to show similar map to Fig. 2 ( $\partial$ pCO2) in Kartadikaria et al (2015).

Most likely your model is overestimating pCO2 in the open ocean region.

**Response**: Noted, we will show similar figure of overall δpCO$_2$ (Difference between pCO$_2$ and atmospheric CO$_2$).

188: that higher => that were higher

**Response**: Thank you for the correction

190-197: There is a significant bias in surface $pCO_2$, especially in the open ocean region surrounding the TMC. This likely explains the much greater carbon outgassing you obtained compared to previous studies.

**Response**: We believe that the referee refers to Figure 3 in the submitted manuscript, where we have compared our simulated $CO_2$ flux with other studies. If that is the case, we would like to clarify that comparison of $CO_2$ flux with previous studies mentioned there was strictly limited to inside the tropical maritime continent region such as the Indonesia seas (95°E-145°E; 10°S-7°N). Although our model still has biases relative to the global reconstruction product, particularly in the open ocean (Landschützer et al., 2016; Iida et al., 2021), these reconstructions did not indicate atmospheric $CO_2$ sink/source characteristics similar to those observed in Kartadikaria et al. (2015) and Hamzah et al. (2020) for the Indonesian seas. However, our model could still reproduce the observed $CO_2$ source signature and encouraged us to further examine its variability.

The difference between reconstruction products and observation-based studies again highlights the issue raised by Lee et al. (2019) concerning the gridded product of data that exhibits substantial inhomogeneity in its sample.

220: I wonder what you consider strong $CO_2$ outgassing. Maybe you could refer to the region(s) with the strongest $CO_2$ outgassing.

**Response**: Thank you for the suggestion, we will revise the sentence in Line 220 by using a more precise criterion.

236: "The biological processes, represented by SSDIC and SSAlk" This is a wrong statement. Changes in sDIC and sAlk are also affected by advection and mixing, and air- sea flux in the case of sDIC. Besides, I would not expect important biology-driven changes in sAlk.

**Response**: Thank you for the explanation, we will revise the explanation

238-239: This is a wrong conclusion based in the wrong assumption that changes in sDIC and sAlk represent biological processes. Consider the upwelling season off Java during summer. sDIC promotes an increase (and sAlk a decrease) in pCO2. Which biological process could explain this? It is not respiration. Most likely, the signature is associated with the upwelling of subsurface waters with higher DIC and alkalinity concentration than the surface waters. During fall, you have a negative impact of sDIC on pCO2, which could reflect a weakening in coastal

upwelling. Remember that in Fig. 5 you are visualizing dpCO2 not pCO2. I would expect a maximum biological uptake of DIC around September. This uptake contributes to decrease sDIC, so its impact should be opposed to the DIC-rich subsurface waters due to upwelling.

**Response**: Thank you for the explanation, we agree on the points you mentioned and we admit our mistake in interpreting the results from $pCO_2$ decomposition analysis. We confirmed that the model result indicates that the maximum uptake of DIC in the area occurs in September just after the wind forcing relaxes from its annual maxima in August, as shown in the figure below.

[Figure]

**Figure AR8. Seasonal cycle of (a) Surface DIC (in μmol kg$^{-1}$); (b) Salinity-normalized DIC (in μmol kg$^{-1}$); and (c) monthly changes in NDIC (in μmol kg$^{-1}$ month$^{-1}$) around South of Java**

Second comment on 238-239, after reading discussion: You mentioned "supply of subsurface inorganic" as a factor impacting $pCO_2$ off Java in the Discussion section (line 339), so I wonder why you did not mention anything of that in the Result section.

**Response**: We have mentioned the high DIC concentration related to upwelling between Line 250-257. We will ensure that in the revised manuscript, the mechanism related to the South of Java upwelling effect on the $pCO_2$ and sea-air $CO_2$ exchange is clearly stated.

268: I wonder why the long-tern negative trend in the fluxes. What does it drive this trend? It may be a model flux drift. Need explanation. Specially if you are highlighting that ENSO and IOD contributed to attenuate this trend.

**Response**: As we have provided previously, the negative trend in the fluxes was the region's response to increasing atmospheric $CO_2$. El Niño and pIOD further induced attenuation in the trend within the interannual timescale.

Figure 6b: It is hard to discriminate the color of the lines. Please increase line width.

**Response**: Noted, we will increase the line width

270: Why do you think it confirms? You are not stating any mechanisms linking the ENSO or IOD variability.

**Response**: The sentence in line 270 was not intended to state any mechanism linking ENSO or IOD. Instead, it was intended to show that the overall $pCO_2$ and sea-air $CO_2$ exchange in the area exhibited modulation most likely related to the recent 2015/16 El Niño and 2019 pIOD. This was based on the period in which $pCO_2$ showed an accelerated trend between 2015/16 and 2019. We can reconsider the sentence whether it is actually necessary or not to put it in the revised manuscript.

285: Why are you using standard deviation instead of the mean index value? I got lost.

**Response**: This was based on the definition of each climatic event. El Niño/La Niña was defined whenever the NINO3.4 SSTA exceeded $\pm 0.5\,^{\circ}C$ for five consecutive months. The IOD on the other hand, defined using standard deviation of DMI (Saji et al. 1999). For convenience, the $CO_2$ flux anomalies and $pCO_2$ anomalies were regressed against a positive one-standard deviation of both NINO3.4 SSTA and DMI. The positive one-standard deviation of both NINO3.4 SSTA and DMI represents typical El Niño and pIOD conditions. We adapted the study by Xiu and Chai (2014) and Pujiana et al. (2019) to conduct this spatial regression analysis.

Figure 8b. I wonder why you did not use a smaller colorbar interval for the flux anomalies.

**Response**: The color bar interval was set to be consistent with Figure 9 in the submitted manuscript, hoping that readers can digest the difference in the magnitude of $CO_2$ flux modulation under ENSO and IOD forcing. Nevertheless, we have provided a similar figure with a smaller color-bar interval at the beginning of this document. We will consider the use of a smaller color bar interval or figure rearrangement if it can boost the readability of the figures.

339: "accelerated gas exchange and an abundant supply of subsurface inorganic". You should try to mention these two processes when describing Figs. 5 & 6 in the Result section.

**Response**: OK. We will add those points in the Result section

350-358: It is not clear to me how the anomalous divergence in the West Pacific affects the air-sea $CO_2$ flux. Could you develop more this idea? I think you need to explain better the counteracting effect of this divergence/convergence with the increased/decreased solar heating during El Nino/La Nina

**Response**: El Niño typically peaked between November and March, which coincides with the northwest monsoon circulation around the TMC. Anomalous atmospheric divergence in the surface caused by descending branch of walker circulation in the western Pacific during El Niño induces anomalous easterly winds in the area, resulting in weaker-than-usual northwest monsoon circulation in some parts of the TMC. As the sea-air gas exchange is proportionately related to the overall wind speed magnitude, we can expect a reduction in the $CO_2$ flux due to decreased wind speed during El Niño, especially in the western Indonesian seas and Southeastern Tropical Indian Ocean.

However, the same atmospheric divergence in the western Pacific also reduced cloud cover in the TMC and increased the amount of downward shortwave radiation (Cai et al., 2019). This results in increased SST and pCO2 and therefore tends to induce positive anomalies in the $CO_2$ flux. The counteracting effect between wind speed, SST, and $\delta pCO_2$ results in relatively smaller magnitudes of $CO_2$ flux anomalies and is even considered insignificant from a statistical perspective in some areas. Please see below figure for more detail on the anomaly distribution associated with the El Niño.

Some areas still showed net positive $CO_2$ flux anomalies during El Niño, such as the South China Sea, and can be traced back to how each component reacts to El Niño (wind speed, $\delta pCO_2,$ and SST anomalies). Generally, wind speed, $\delta pCO_2$, and SST anomalies in the South China Sea exhibit an in-phase relationship with El Niño.

The La Niña on the opposite, induces stronger-than-usual northwest monsoon circulation due to the anomalous atmospheric convergence in near surface. This will lead to accelerated gas exchange. However, increased cloud cover in the western Pacific, including the Tropical Maritime Continent reduces the downward shortwave radiation and decrease the SST in some part of the area. Again, counteracting effect between the atmosphere side and ocean side during La Niña also results in relatively smaller magnitudes of $CO_2$ flux anomalies.

The IOD on the other hand, typically occurs between July and November, which coincides with the upwelling season in South Java. During the positive IOD (pIOD) event, increased wind speed around the Southeastern Tropical Indian Ocean during the pIOD not only accelerates the gas exchange but also enhances the upwelling strength (Delman et al., 2016; Horii et al., 2018). The combination of accelerated gas exchange and enhanced upwelling

during the pIOD in South Java resulted in strong anomalous $CO_2$ degassing from the area. The opposite pattern occurs during negative IOD where decreased wind speed led to weakened upwelling and ultimately results in anomalously weak $CO_2$ degassing.

[Figure]

**Figure AR9. Regressed (a) wind speed (vector arrows) and magnitude (shaded color), all in m s$^{-1}$ units; (b) anomalies in the difference between sea surface pCO$_2$ and atmospheric CO$_2$ (δpCO$_2$, in μatm); and (c) SST anomalies (in °C) against a one-standard deviation of NINO3.4 SSTA (+1σ$_{NINO3.4}$) representing typical El Niño events according to the HIS scenario results. Plotted vector and shaded color were significant at $p > 0.01$**

[Figure]

**Figure AR10. Same as Figure 8 but for Positive IOD case using one-standard deviation of DMI (+1σ_DMI)**

**References**:

Cai, W., Wu, L., Lengaigne, M., Li, T., McGregor, S., Kug, J. S., Yu, J. Y., Stuecker, M. F., Santoso, A., Li, X., Ham, Y. G., Chikamoto, Y., Ng, B., McPhaden, M. J., Du, Y., Dommenget, D., Jia, F., Kajtar, J. B., Keenlyside, N., Lin, X., Luo, J. J., Martín-Rey, M., Ruprich-Robert, Y., Wang, G., Xie, S. P., Yang, Y., Kang, S. M., Choi, J. Y., Gan, B., Kim, G. Il, Kim, C. E., Kim, S., Kim, J. H., and Chang, P.: Pantropical climate interactions, Science (80-. )., 363, https://doi.org/10.1126/science.aav4236, 2019.

Delman, A. S., Sprintall, J., McClean, J. L., and Talley, L. D.: Anomalous Java cooling at the initiation of positive Indian Ocean Dipole events, J. Geophys. Res. Ocean., https://doi.org/10.1002/2016JC011635, 2016.

Hamzah, F., Agustiadi, T., Susanto, R. D., Wei, Z., Guo, L., Cao, Z., and Dai, M.: Dynamics of the Carbonate System in the Western Indonesian Seas During the Southeast Monsoon, J. Geophys. Res. Ocean., 125, 1–18, https://doi.org/10.1029/2018JC014912, 2020.

Horii, T., Ueki, I., and Ando, K.: Coastal upwelling events along the southern coast of Java during the 2008 positive Indian Ocean Dipole, J. Oceanogr., 74, 499–508, https://doi.org/10.1007/s10872-018-0475-z, 2018.

Key, R. M., Kozyr, A., Sabine, C. L., Lee, K., Wanninkhof, R., Bullister, J. L., Feely, R. A., Millero, F. J., Mordy, C., and Peng, T. H.: A global ocean carbon climatology: Results from Global Data Analysis Project (GLODAP), Global Biogeochem. Cycles, 18, 1–23, https://doi.org/10.1029/2004GB002247, 2004.

Kartadikaria, A. R., Watanabe, A., Nadaoka, K., Adi, N. S., Prayitno, H. B., Soemorumekso, S., Muchtar, M., Triyulianti, I., Setiawan, A., Suratno, S., and Khasanah, E. N.: CO2 sink/source characteristics in the tropical Indonesian seas, J. Geophys. Res. Ocean., 120, 7842–7856, https://doi.org/10.1002/2015JC010925,      2015.

Landschützer, P., Gruber, N., and Bakker, D. C. E.: Decadal variations and trends of the global ocean carbon sink, Global Biogeochem. Cycles, 30, 1396–1417, https://doi.org/10.1002/2015GB005359, 2016.

Lee, T., Fournier, S., Gordon, A. L., and Sprintall, J.: Maritime Continent water cycle regulates low-latitude chokepoint of global ocean circulation, Nat. Commun., 10, 1–13, https://doi.org/10.1038/s41467-019-10109-z, 2019.

Locarnini, R. A., A. V. Mishonov, O. K. Baranova, T. P. Boyer, M. M. Zweng, H. E. Garcia, J. R. Reagan, D. Seidov, K. Weathers, C. R. Paver, and I. Smolyar, 2018. World Ocean Atlas 2018, Volume 1: Temperature. A. Mishonov Technical Ed.; NOAA Atlas NESDIS 81, 52pp.

Pujiana, K., McPhaden, M. J., Gordon, A. L., and Napitu, A. M.: Unprecedented Response of Indonesian Throughflow to Anomalous Indo-Pacific Climatic Forcing in 2016, J. Geophys. Res. Ocean., 124, 3737–3754, https://doi.org/10.1029/2018JC014574, 2019.

Saji, N. H., Goswami, P. N., Vinayachandran, P. N., and Yamagata, T.: Saji,N.A et al,.dipole mode in the tropical indain ocean, Nature, 401, 360–363, 1999.

Xiu, P. and Chai, F.: Variability of oceanic carbon cycle in the North Pacific from seasonal to decadal scales, J. Geophys. Res. Ocean., 119, 5270–5288, https://doi.org/10.1002/2013jc009505, 2014.

Zweng, M. M., J. R. Reagan, D. Seidov, T. P. Boyer, R. A. Locarnini, H. E. Garcia, A. V. Mishonov, O. K. Baranova, K. Weathers, C. R. Paver, and I. Smolyar, 2018. World Ocean Atlas 2018, Volume 2: Salinity. A. Mishonov Technical Ed.; NOAA Atlas NESDIS 82, 50pp.

---

## Author Comment (AC2)

Response to Referee #1 Comment (RC1)

We would like to thank the referee for taking the time to review our manuscript and for providing constructive comments. Please see our responses to the comments below.

The paper is a long-standing pending issue in addressing the CO2 fluxes of this very important region. It deserves publication in Ocean Science. However, it has missed the following key elements, which need to be addressed.

Valsala et al., (2020) have done, for the first time, analysis of 60-year long record of sotheastern tropical Indian Ocean CO2 flux variability, pCO2 variability associated with the IOD. The study concluded that, "The IOD leads to a substantial sea-to-air CO2 flux variability in the southeastern tropical Indian Ocean over a broad region (70–105°E, 0–20°S), with more focus near the coast of Java-Sumatra due to the prevailing upwelling dynamics and associated westward propagating anomalies. The sea-to-air CO2 fluxes, surface ocean partial pressure of CO2 (pCO2), the concentration of dissolved inorganic carbon (DIC), and ocean alkalinity (ALK) range as much as ±1.0 mole m−2 year−1, ±20 μatm, ±35 μmole kg−1, and ±22 μmole kg−1 within 80–105°E, 0–10°S due to IOD. The DIC and ALK are significant drivers of pCO2 variability associated with IOD. The roles of temperature (T) and biology are found negligible. A relatively warm T and extremely high freshwater forcing make the southeastern tropical Indian Ocean carbon cycle variability submissive to DIC and ALK evolutions in contrast to the tropical eastern Pacific where changes in DIC and T dominate the pCO2 interannual variability. For the first time, this study provides a most comprehensive and extended analysis for the region while highlighting significant differences in carbon cycle variability of the eastern tropical Indian Ocean compared to that of the other parts of the global oceans."

This is an important recent work and needs to be cross-discussed in this paper, especially due to the reason the IOD impacts are revisited in this manuscript, and the results have differences. It is always good to have various modelling comparisons so that the community is benefited from knowing how the model performs and differ from each other. Other papers missed addressing are also added below:

Valsala, V., M. G. Sreeush, and K. Chakraborty, (2020), IOD impacts on Indian the Ocean Carbon Cycle, Journal of Geophysical Research, https://doi.org/10.1029/2020JC016485

Chakraborty K., V. Valsala, T. Bhattacharya, J. Ghosh, (2021), Seasonal cycle of surface ocean pCO2 and pH in the northern Indian Ocean and their controlling factors, Progress in Oceanography, Vol.198, doi.org/10.1016/j.pocean.2021.102683

Valsala V., Sreeush M.G., Anju M., Sreenivas P., Tiwari Y.K., Chakraborty K., Sijikumar S., An observing system simulation experiment for Indian Ocean surface pCO2 measurements, Progress in Oceanography, 194: 102570, June 2021, DOI: 10.1016/j.pocean.2021.102570, 1-14

Sreeush, M. G., Valsala, V., Pentakota, S., Prasad, K. V. S. R., and Murtugudde, R (2018), Biological production in the Indian Ocean upwelling zones – Part 1: refined estimation via the use of a variable compensation depth in ocean carbon models, Biogeosciences, 15, 1895-1918, https://doi.org/10.5194/bg-15-1895-2018

**Response**: Thank you very much for your comment and for providing us with a recent study that is relevant to our manuscript. We will add some discussion, especially regarding the previous modeling results, in the revised manuscript. For a lively community discussion here, we will explain some key points we obtained from the articles provided by the referee.

Both our model and the study by Valsala et al. (2020) agree that a positive IOD (pIOD) corresponds with stronger-than-usual sea-air $CO_2$ flux in the southeastern tropical Indian Ocean (SETIO). The important role of DIC which outweigh the temperature influence on regulating the $pCO_2$ and further $CO_2$ flux in the area also highlighted from these modeling studies. However, we also noted some notable differences between this modeling study and the recent study by Valsala et al. (2020).

There is a difference regarding the center of IOD influence on sea-air $CO_2$ flux modulation suggested in Valsala et al. (2020) compared with our modeling results. Instead of concentrated IOD influence on the southwestern part of the Sumatra coast, study conducted by us suggested another strong $CO_2$ flux modulation signal on the south coast of Java, even with limited local wind modulation under the same climatic forcing. We suppose that this pattern is related to the influence of remote forcing, as indicated by Delman et al. (2016, 2018). They suggested that anomalous wind stress west of Sumatra or equatorially forced Kelvin Wave plays an important role in upwelling variabilities in South Java during pIOD events. Because the model utilized here was forced by high-temporal resolution atmospheric data (Three-hourly JRA55; Kobayashi et al., 2015) and all the tracers (physical and biogeochemical) were calculated online during model integration, it is likely that our model captured the intraseasonal

variabilities during IOD development. The intraseasonal signal accumulated further and resulted in an interannual pattern, as shown in Figure 9 in the submitted manuscript.

To further prove that the regressed $pCO_2$ and $CO_2$ flux anomaly patterns in the submitted manuscript are related to the IOD-induced upwelling variabilities, we also regressed the wind speed anomalies, anomalies in the difference between sea surface $pCO_2$ and atmospheric $CO_2$ ($\delta pCO_2$), and SST anomalies against a one-standard deviation of the DMI, as shown in figure below. The regressed patterns of wind anomalies and SST anomalies were consistent with the study by Delman et al. (2016), who utilized satellite data. Therefore, it is very likely that the presented $pCO_2$ and $CO_2$ flux anomaly pattern related to the IOD is due to upwelling modulation, especially along the southern Sumatra-Java coast.

[Figure]

**Figure AR1. Regressed (a) wind speed (vector arrows) and magnitude (shaded color), all in m s$^{-1}$ units; (b) anomalies in the difference between sea surface pCO$_2$ and atmospheric CO$_2$ ($\delta$pCO$_2$, in $\mu$atm); and (c) SST anomalies (in $^\circ$C) against a one-standard deviation of DMI (+1$\sigma_{DMI}$), representing typical positive IOD events according to the simulation results that use historical monthly atmospheric CO$_2$ concentrations. Plotted vector and shaded color were significant at $p > 0.01$**

Another point that should be noted is the possible effect of the horizontal resolution setting used in our model (i.e., $1/6^\circ \times 1/6^\circ$), which may limit the extent of IOD influence related to upwelling variabilities. A recent study by Kido et al. (2022) argued that low-resolution OGCM exaggerated the coastal-open ocean water exchange artificially, which is in line with the experiments conducted by Liu et al. (2019). Additionally, Delman et al. (2018) in their study also indicated that a relatively coarse resolution model product underestimated the advection effect (both horizontal and vertical) on South Java cooling during the development of the pIOD. This may explain the relatively smaller extent of IOD influence according to our simulation results relative to Valsala et al. (2020).

**Reference:**

Delman, A. S., McClean, J. L., Sprintall, J., Talley, L. D., and Bryan, F. O.: Process-Specific Contributions to Anomalous Java Mixed Layer Cooling During Positive IOD Events, J. Geophys. Res. Ocean., 123, 4153–4176, https://doi.org/10.1029/2017JC013749, 2018.

Delman, A. S., Sprintall, J., McClean, J. L., and Talley, L. D.: Anomalous Java cooling at the initiation of positive Indian Ocean Dipole events, J. Geophys. Res. Ocean., https://doi.org/10.1002/2016JC011635, 2016.

Kido, S., Nonaka, M., and Miyazawa, Y.: JCOPE-FGO: an eddy-resolving quasi-global ocean reanalysis product, Ocean Dyn., 72, 599–619, https://doi.org/10.1007/s10236-022-01521-z, 2022.

Liu, X., Dunne, J. P., Stock, C. A., Harrison, M. J., Adcroft, A., and Resplandy, L.: Simulating Water Residence Time in the Coastal Ocean: A Global Perspective, Geophys. Res. Lett., 46, 13910–13919, https://doi.org/10.1029/2019GL085097, 2019.

Kobayashi, S., Ota, Y., Harada, Y., Ebita, A., Moriya, M., Onoda, H., Onogi, K., Kamahori, H., Kobayashi, C., Endo, H., Miyaoka, K., and Kiyotoshi, T.: The JRA-55 reanalysis: General specifications and basic characteristics, J. Meteorol. Soc. Japan, 93, 5–48, https://doi.org/10.2151/jmsj.2015-001, 2015.

Valsala, V., Sreeush, M. G., and Chakraborty, K.: The IOD Impacts on the Indian Ocean Carbon Cycle, J. Geophys. Res. Ocean., 125, 1–18, https://doi.org/10.1029/2020JC016485, 2020.